# Exploring context for implementation of inclusive education for children with developmental disabilities in mainstream primary schools in Ethiopia

Elisa Genovesi[1]*, Ikram Ahmed[2], Moges Ayele[3], Winini Belay[4], Olivia Burningham[5], Amanda Chen[6], Fikirte Girma[2], Liya Tesfaye Lakew[7], Charlotte Hanlon[8‡], Rosa Anna Hoekstra[1‡]

1 Department of Psychology, Institute of Psychiatry, Psychology & Neuroscience, King's College London, London, United Kingdom, 2 Department of Psychiatry, WHO Collaborating Centre for Mental Health Research and Capacity-Building, School of Medicine, College of Health Sciences, Addis Ababa University, Addis Ababa, Ethiopia, 3 School of Psychology, College of Education and Behavioral Studies, Addis Ababa University, Addis Ababa, Ethiopia, 4 Centre for Innovative Drug Development and Therapeutic Trials for Africa, Addis Ababa University, Addis Ababa, Ethiopia, 5 Department of International Development, London School of Economics and Political Science, London, United Kingdom, 6 St Andrew's Mission School, Singapore, Singapore, 7 Nia Foundation Joy Center for Autism, Addis Ababa, Ethiopia, 8 Centre for Clinical Brain Sciences, Deanery of Clinical Sciences, University of Edinburgh, Edinburgh, United Kingdom

‡ CH and RAH are senior authors on this work
* elisa.genovesi@kcl.ac.uk

**Data Availability Statement:** The data supporting this article is not openly available as participants did not give explicit consent to archiving and may

## Abstract

A large gap in provision of services for children with developmental disabilities (DD) has been identified in Ethiopia, especially in the education system. Including children with disabilities in mainstream schools is encouraged by policies, but progress in this direction has been limited. This study aimed to explore stakeholders' perspectives on contextual factors relevant for inclusive education for children with DD in mainstream schools in Ethiopia, with a focus on Adis Ababa. Data were collected through semi-structured interviews with 39 local stakeholders, comprising caregivers of children with DD, school teachers and principals/managers, non-governmental organisation representatives, government officials, clinicians and academics/consultants. We used template analysis to code the data and map them onto domains of the Context and Setting dimensions of the *Context and Implementation of Complex Interventions* framework. Stakeholders discussed frameworks in the Legal and Ethical context endorsing the right of all children to education. However, they reported multiple reasons why children with DD in Ethiopia have limited access to education, either in special or mainstream schools. First, individual features, such as gender and support needs, discussed in the Epidemiological context, may affect the likelihood of a child with DD to be accepted in school. Transportation challenges are a key barrier in the Geographical context. Socio-economic and Socio-cultural contexts present barriers at the levels of the nation, school and family, mostly related to limited services and material and financial resources and limited awareness of DD. Stakeholders believe the currently limited but growing commitment in the Political context can support progress towards the removal of these barriers.

be recognisable from interview transcripts. It may be shared on request for academic or clinical research with ethical approval. A descriptive record can be found in the King's College London research data repository at https://doi.org/10.18742/25443721. To request access for academic or clinical research with ethical approval, email research.data@kcl.ac.uk.

**Funding:** The project has been funded by the UKRI Economic and Social Research Council through a London Interdisciplinary Social Science Doctoral Training Partnership (LISS DTP) studentship (ES/P000703/1), by the British Institute for Eastern Africa through a Thematic Grant and by undergraduate research project funding from King's College London. RAH, CH, FG and WB receive support from the National Institute for Health and care Research (NIHR200842) using UK aid from the UK Government and CH through the NIHR Global Health Research Group on Homelessness and Mental Health in Africa (NIHR134325) using UK aid from the UK Government. The views expressed in this publication are those of the authors and not necessarily those of the NIHR or the Department of Health and Social Care or other funders. CH receives support from the Wellcome Trust through grants 222154/Z20/Z and 223615/Z/21/Z.

**Competing interests:** The authors have declared that no competing interests exist.

Our findings can form the basis for development of an implementation plan that addresses such barriers and capitalises on existing facilitators.

## Introduction

Developmental disabilities (DD), including autism and intellectual disability, are characterised by difficulties or differences in cognitive ability, communication and social behaviour beginning in childhood [1]. They are a substantial cause of disability globally and characterised by high support needs [2, 3]. Children with DD may need a range of healthcare and educational services to support their cognitive, social and motor development, including the assistance of mental health and rehabilitation specialists. However, these services are often limited, particularly in low- and middle-income countries, where most children with DD reside [3].

In Ethiopia, a very low-income country in Eastern Africa, diagnostic, healthcare, educational and rehabilitation services for children with DD are limited [4] and often unknown to families who need them [5]. The limited availability and access to services are aggravated by stigma towards children with DD [6–8], which is largely unaddressed [5] and often leads to the social isolation of these children and their caregivers [7]. In a study surveying 102 caregivers of children with DD in the capital city, Addis Ababa, stigma was reported by the majority of respondents [8]. Additionally, 47.1% of caregivers reported unmet needs in receiving professional support for their children with DD. The most prominent unmet needs of caregivers, reported by 74.5% of respondents, was access to appropriate education [8]. In more recent qualitative studies, caregivers spoke of their wish for their children to be schooled [9, 10]. While promoting child development [11], inclusion in education can also ensure that caring responsibilities are not placed exclusively on caregivers, as is often the case in Ethiopia [6]. Based on a situational analysis of services for DD in the country [4], a limited number of children with DD are enrolled in the few special schools located in Addis Ababa. A few inclusive education programmes are present in some mainstream schools [4]. However, these schools and programmes have long waiting lists and are not sufficient to accommodate all children with DD, who are often rejected from mainstream schools [4].

The most recent Ethiopian Education Sector Development Programme [12] reported that education was provided to only 11% of all children with any special needs, although data specifically pertaining to children with DD were missing. The programme promotes inclusive education, also indicated by the governments endorsement of the United Nations (UN) Convention on the Rights of Persons with Disabilities (CRPD) [13]. However, progress towards inclusive education has proved difficult, mainly due to challenges in providing mainstream school teachers with appropriate resources and training [14, 15]. Similar barriers were reported in a recent situational analysis of inclusive education in Ethiopia [16], commissioned by UNESCO in preparation of the 2020 Global Education Monitoring Report [17], in qualitative and mixed-methods studies exploring teachers' perspectives in Addis Ababa [18], the northern region of Tigray [19] and southern Ethiopia [20].

These studies all considered children with any disabilities, rather than children with DD in particular. The complex needs of children with DD mean that they may experience higher exclusion rates than children with physical or sensory disabilities. For example, in Uganda, children with cerebral palsy and associated intellectual disability were four times less likely to attend school compared to those with cerebral palsy but no intellectual disability [21]. Further, a recent systematic review of 32 qualitative studies in sub-Saharan Africa [22], highlighted that

in multiple countries the few available special needs education training programmes for teachers gave them little or no information on how to address the needs of children with DD. In Ethiopia, the most recent "Master Plan" for the education of children with disabilities [23] focuses strongly on provisions and modifications for children with sensory and physical disabilities (e.g. teaching materials in Braille, sign language training for staff, ramps to access facilities, etc): relevant accommodations for children with DD are less prominent. Therefore, it is likely that the inclusion and quality education of children with DD in Ethiopia may be hindered by greater challenges than those reported in general disability studies [18–20].

A full picture of contextual factors and an understanding of local stakeholder perspectives on inclusive education is crucial so that strategies can be identified to improve educational access for children with DD in Ethiopia. In this study, we interviewed Ethiopian stakeholders to explore their perspectives on including children with DD in education in Ethiopia and particularly the capital city, Addis Ababa, with the goal to ultimately develop and implement a context-appropriate intervention. In a previous in-depth analysis of these data we focused on the potential benefits and drawbacks of inclusive education specifically for the development and inclusion iof children with DD [11]. We concluded that inclusive education can benefit children with DD, if potential drawbacks are managed, and as such is considered an acceptable goal to pursue. In the current report we present a comprehensive analysis of the same data, focusing not primarily on the children themselves, but on contextual factors relevant to the inclusion of children with DD in primary schools in Addis Ababa. This report and a companion paper reporting on implementation processes [24] together lead to a context-appropriate framework-based implementation model for educational inclusion of children with DD.

Stakeholders should be involved in all stages of intervention development and implementation [25]. One key contribution they can give is in identifying relevant contextual factors that can influence implementation: from macro-level factors—national regional; to meso-level factors—in communities and organisations, for example the school community; to micro-level factors—where interpersonal interactions occur, for example in families or between teacher and children [25]. As such, the views of stakeholders involved in the care of children with DD and in education provision in Ethiopia can shed light on contextual barriers and facilitators for promoting the education of children with DD in this context.

Specifically, in the analysis reported below, we investigated stakeholders' perspectives on the existing education needs and challenges of children with DD in Ethiopia, with a focus on Addis Ababa, and on contextual and attitudinal barriers and facilitators for their inclusion in mainstream primary schools.

## Materials and methods

We conducted a qualitative study comprising semi-structured interviews with key stakeholders with expertise in DD and/or education in Addis Ababa, the capital city of Ethiopia, from April 2022 to February 2023.

### Participants

We planned to interview 8–10 caregivers, 10–15 school representatives (teachers and principals/managers) and 15–10 other professional informants (Non-governmental organisation—NGO—representatives, academics, clinicians, government officials), to ensure a range of roles and perspectives for each participant group.

Caregivers and teachers were recruited from special and mainstream schools using purposive sampling, aiming to ensure that interviewees included caregivers whose children received education in special centers, special units and inclusive classes and, similarly, teachers working

in a variety of settings. Caregivers were also recruited from healthcare clinics, with the purpose of seeking caregivers whose children did not attend school. Administration offices at schools and clinics selected by convenience were first contacted as gatekeepers. The informed consent and interviews were conducted by researchers independent of the school administration to ensure voluntary participation. Other stakeholders were recruited through convenience and snowballing sampling, starting by directly contacting stakeholders known to the research team, and subsequently those recommended by interviewees.

## Data collection

Eighteen semi-structured interviews were conducted in English and 21 in Amharic. All caregivers and teachers were interviewed in Amharic, exept for a clinician interviewed in English who is also a primary caregiver. Five interviews in English were conducted virtually, either via phone call or video call, due to practical constraints, while all others happened in person in Addis Ababa.

Interviews consisted of a conversation guided by flexible questions on the meaning of, and value placed on, inclusive education for children with DD, their current educational situation and barriers to education access and quality, and routes to improvement. The interview guide (S1 File) was developed by the first author to address key aspects of the research question and was iteratively revised by the research team and following a trial interview with an Ethiopian clinician. It presented variations depending on participant categories, to ensure that everyone's expertise and experience was elicited: for example, caregivers were specifically asked about their child's needs and education experience and teachers about experiences in the classroom. Interview durations varied from 20 minutes to two hours; most interviews lasted 35–55 minutes.

All interviews were audio-recorded and transcribed verbatim, to ensure fidelity to the audio version in the transcription. Transcripts in Amharic were translated to English by an external party.

Demographic data were collected through a brief questionnaire.

## Ethical approval

Ethical approval was obtained from King's College London (RESCM-22/23-21930), the Scientific and Ethical Review Committee at CDT-Africa of Addis Ababa University and the Addis Ababa University College of Health Science Institutional Review Board (061/21/CDT). All participants provided written informed consent to participate.

## Analysis

Interview transcripts were analysed using thematic template analysis [26], an approach that is compatible with requirements of the current project: first, co-development of themes in the research team, to ensure multidisciplinary understanding of the data, and Ethiopian as well as outsider perspectives; secondly, the use of an implementation framework to provide structured information for the development and implementation of an intervention.

Four researchers (EG, WB, AC, OB) familiarised themselves with the interview data by transcribing them or reading transcripts, then iteratively developed inductive codes on meaningful sections of the transcripts in NVivo. Each coder independently analysed a subset of the data, after consensus meetings based on initial coding of a few interviews. Once codes were developed for and applied to all interviews, EG organised them deductively in a thematic template based on the Context and Setting dimensions of the Context and Implementation of Complex Interventions (CICI) framework [25]. Throughout these phases, whole-team

meetings were held to discuss interpretations and theme development. Participatory stakeholder validation of the analysis was sought by presenting intial themes on contextual factors and eliciting discussion on them at three workshops attended by a variety of stakeholders.

The CICI was chosen for its comprehensive consideration of context, synthesising several previous frameworks, as well as its consideration for how different contextual factors can interact with and affect intervention implementation [25]. Contextual factors are considered at the macro, meso and micro levels and categorised into multiple domains [25]. The geographical context comprises natural and infrastructural environments surrounding implementation. The epidemiological context includes specific features of the target group (in this case, children with DD) and demographics associated with different needs. The socio-economic context concerns the availability of and access to resources and services. The socio-cultural context includes beliefs, common knowledge, patterns of behaviours, cultural customs. The political context describes the distribution of power and the interests of different bodies and individuals holding power. The legal context comprises established laws and rules. The ethical context concerns moral values and principles. Finally, the Setting is a separate CICI dimension, referring to the physical location where an intervention is implemented.

Within themes/domains, subthemes were created inductively from codes, and mapped to macro (whole-country, wider services), meso (within school settings), and micro (relative to the class setting, families or individuals) context levels when relevant to the data. In the Results section, the analysis is exemplified with interview quotes. The first letter of participants' codes refers to the language used in the interview ("E" for English, "A" for Amharic).

## Positionality

The philosophical stance of the study is critical realism [27]: the researchers aimed to investigate an existing reality, and acknowledged that this can only be explored through the different perspectives of stakeholders. In turn, stakeholders' perspectives are interpreted by researchers through their own subjectivity. Consequently, researchers were reflexive of their positionality throughout the study [28].

The research team comprises five non-Ethiopian UK-based researchers and students (one based in Ethiopia) and five Ethiopian investigators, including a researcher, a caregiver and special school representative, an academic developmental psychologist, and two clinical academics. All believed in the importance of increasing access to quality education for children with DD.

Of the four researchers who lead the analysis, one is Ethiopian and three identify as outsiders in relation to the Ethiopian population. As such, reflexivity on personal assumptions, and coding discussions between non-Ethiopian and Ethiopian team members were crucial to ground data interpretation in the Ethiopian context.

## Results

### Participants

We interviewed 39 Ethiopian adult stakeholders (20 women), aged from 28 to 65 years. Numbers and gender for each participant category are reported in Table 1.

The majority of caregivers were housewives, except for a trader and a practicing clinician. Five were married, 2 were single, 2 divorced and 1 widowed. Nine were mothers of the child with DD they cared for and one was an aunt. The ages of the children with DD (4 boys, 6 girls) ranged from 8 to 24, as no limits were imposed when recruiting caregivers, and their diagnoses are shown in Table 2.

**Table 1. Participant types and gender.**

| Participant type | Number of male participants | Number of female participants |
|---|---|---|
| Teachers (mainstream or special education) | 5 | 3 |
| Public school principals | 1 | 0 |
| Managers/heads/representitives of private schools (inclusive or special) | 2 | 2 |
| Primary caregivers | 0 | 10* |
| Practicing clinicians (pediatricians, clinical psychologists, psychiatrists) | 1 | 6* |
| Government officials (local and national) | 3 | 0 |
| Academics/consultants | 3 | 0 |
| NGO representatives | 4 | 0 |

*One female participant was both a practicing clinician and a primary caregiver and has been reported in both categories

## Analysis

Table 3 reports the thematic template developed. The full summary of codes relative to each theme (codebook) is available in S2 File.

**Theme 1: Geographical context.**   Discussions on geographical context revolved around infrastructure: specifically, transport and the geographical accessibility of education services for children with DD.

As very few schools (in both the mainstream and special education sector) accept children with DD, they are out of geographical reach for most families. This problem is particularly pronounced for families living in rural areas, where local schools typically do not accept children with DD, due to poorer infrastructure and more negative attitudes towards disabilities. Therefore, these families have to seek enrollment for their children in cities. Even within Addis Ababa, distance creates significant barriers when paired with many challenges in accessing transport.

A024: You can't trust the taxi [minibus, main mean of public transport] and Bajaj [three-wheel taxi] drivers to take her to school, and they are expensive. (caregiver)

**Theme 2: Epidemiological context.**   *Subtheme 2.1*: *Diversity of needs*. Stakeholders highlighted how children with DD are all different, with a range of support needs, which may influence the choice of education setting. Several needs were discussed, including learning difficulties in intellectual and adaptive skills, aggression and other harmful behaviours, hyperactivity, sensory sensitivity needs, communication and social difficulties, and comorbid physical and motor disabilities or medical conditions. Caregivers also reported the different strengths of their children.

A031: Her strengths are that she loves music and works hard. (caregiver)

**Table 2. Diagnoses of caregiver participants' children.**

| Diagnosis | Number of children |
|---|---|
| Autism only | 5 |
| Autism with coexisting epilepsy and intellectual disability | 2 |
| Cerebral palsy | 1 |
| Attention-deficit hyperactivity disorder | 2 |

**Table 3. Thematic template.**

| |
|---|
| Theme 1: Geographical context |
| Theme 2: Epidemiological context |
| Subtheme 2.1: Diversity of needs |
| Subtheme 2.2: Gender and DD |
| Theme 3: Socio-economic context |
| Subtheme 3.1: Macro socio-economic context |
| Subtheme 3.2: Meso socio-economic context |
| Subtheme 3.3: Micro socio-economic context |
| Subtheme 3.3.1: Access to individual support |
| Subtheme 3.3.2: Family resources |
| Theme 4: Socio-cultural context |
| Subtheme 4.1: Macro socio-cultural context |
| Subtheme 4.2: Meso socio-cultural context |
| Subtheme 4.3: Micro socio-cultural context |
| Subtheme 4.3.1: Interpersonal class context |
| Subtheme 4.3.2: Family socio-cultural context |
| Theme 5: Political context |
| Subtheme 5.1: Macro political context |
| Subtheme 5.2: Meso political context |
| Theme 6: Legal context |
| Subtheme 6.1: Macro legal context |
| Subtheme 6.2: Meso legal context |
| Theme 7: Ethical context |
| Theme 8: Setting |

Multiple professionals and caregivers highlighted the importance of recognising the potential of children with DD: from government official A030 who suggested that, depending on their support needs, children with DD could "become nurses or soldiers", make objects such as "mats and chairs" or "do cleaning jobs", to special school teacher A033, who told the story of the student who "went to Dubai and brought a trophy".

*Subtheme 2.2*: *Gender and DD*. Interviewers often prompted participants to consider gender influences on inclusion, to account for intersectional exclusion and guide the development of interventions serving boys and girls equally.

Most participants thought boys and girls with DD faced the same challenges in accessing education. A few noted barriers that could disproportionately affect girls. One reported barrier was the inherent exclusion of girls, regardless of disability:

E005: So there is a belief that if a girl is sent to school, she would reach [success] and know her name or... There is a belief in a community like that she couldn't. She can't [have] success in school. She can't reach. She can't go out. She can't work. She can't. For a girl that's general.

Secondly, as the system is more attuned to identify male children with DD and meet their needs, stakeholders reported that girls with DD may not receive the necessary services and accommodations. Thirdly, vulnerability to sexual abuse was a recurrent concern for children with DD in inclusive schools and was reported to disproportionally affect girls:

E013: One of the negative issue is this abuse and the like. The girls could be more affected with that.

**Theme 3: Socio-economic context.** *Subtheme 3.1*: *Macro socio-economic context*. Informants considered Ethiopia's status as a very low-income country where "all of the projects are donor-driven" (A019, academic) a key contextual factor when considering the inclusion of children with DD in education. An overall lack of resources and services was highlighted. At the same time, stakeholders recognised the valuable role of NGOs and UN organisations and their collaboration with the Ethiopian government.

A036: I think NGOs can do a lot to support inclusive education. They can help children with DD with many things. (teacher)

Beyond the limited access to services, including identification, education, healthcare and employment support services, stakeholders reported a poverty of knowledge resources, specifically of research developments and of appropriate formal training for teachers and special needs education professionals. However, some progress was reported in all these areas.

A019: Believe it or not, we didn't have any courses on autism until last year. [. . .] The pioneering department in special needs education established it at Addis Ababa University only last year. (academic)

*Subtheme 3.2*: *Meso socio-economic context*. The lack of economic resources translated into a lack of material and human resources on the ground in school settings. Some private and special schools have resources to provide quality support, but this is only accessible to a limited number of children, generating long waiting lists. At the same time, in government schools, teachers reported not feeling able to address the needs of a child with DD in mainstream classes. They lack special education materials, assistive devices and, most importantly, assistants, as a class of 50–80 students was typically under the responsibility of only one teacher at any given time. Children with DD are therefore turned away, more so than children with physical and sensory disabilities, perceived as having lower support needs. In government schools that accept children with DD, these are cared for in a special unit by a small number of special needs education teachers. Moreover, this profession in under-incentivised and affected by high turn-over.

E006: Being a special need expert or special need teacher, that needs interest, that is commitment and that needs to take everything towards to use. So many teachers are not encouraged to do such things. (government official)

*Subtheme 3.3*: *Micro socio-economic context*. **Subtheme 3.3.1**: **Access to individual support.** When children with DD access under-resourced government school settings, they often receive limited support in class, particularly in mainstream classes. Stakeholders discussed how this negatively affects their progress.

E011: He is included in the classrooms, but nobody is assisting him based on his needs. (clinician)

**Subtheme 3.3.2**: **Family resources.** The overall lack of services also impacts families of children with DD. First, caregivers, often single mothers isolated by their families due to stigma, are unsupported in caring for their children who require constant supervision. In some cases, access to school helps, although it may also create further strain, especially when accompanying the child to school for long distances.

A026: If it had not been for his teachers, I would have had to watch him without doing any work. (caregiver)

A024: It is tiresome. You have to wait in line to get a taxi [public minibus]. It takes me a long to get back home (caregiver)

Secondly, striving to access services for their children generates expenses: paying for access in private well-resourced healthcare and education facilities, buying materials, foregoing work to assist the child in class or hire an assistant.

Overall, stakeholders identified a need to support caregivers financially and psychologically, as well as helping them in tasks such as caring for their children and enrolling them in school.

**Theme 4: Socio-cultural context.** According to stakeholders, social and cultural perspectives at macro, meso and micro levels are mostly characterised by limited awareness about DD and on the needs, rights and strengths of children with DD.

*Subtheme 4.1*: *Macro socio-cultural context*. Awareness has spread in the past decade, including through the work of Zemi Yenus, founder of Ethiopia's first school for autistic children, the Joy Center for autism.

A031: Zemi is not alive now, but she has left behind a legacy. Public awareness of autism has increased a lot. (caregiver)

Nonetheless, the lack of awareness in the community and among professionals was still seen as a barrier to identification and recognition, inclusion in society and access to health and education services. Limited awareness took many forms, including spiritual explanations of DD and negative attitudes towards children with DD. Most of all, stakeholders stressed the need to raise awareness "that these children can learn new things, new skills" (E001, clinician) and "that these children can work" (A032, caregiver).

*Subtheme 4.2*: *Meso socio-cultural context*. Stakeholders identified similar attitudinal barriers within schools, reporting that some principals and teachers display negative attitudes towards children with DD or lack understanding of their needs and potential.

A029: Teachers could punish them because they lack awareness of their condition. [. . .] They could consider them lazy. (government official)

Additionally, parents of typically developing children can discourage the inclusion of children with DD, by complaining to teachers and principals.

E004: Some parents would come in "I don't want you to put my children/child in that child's class", or "don't allow him to get next to that child with disability or special needs". (private inclusive school head)

*Subtheme 4.3*: *Micro socio-cultural context*. **Subtheme 4.3.1: Interpersonal class context.** Negative beliefs and attitudes were also reported among typically developing peers, causing barriers to inclusion within mainstream classes. Relationships with peers were in fact one of the main concerns relative to the inclusion of children with DD in mainstream schools. Specifically, stakeholders reported worries that children with DD may get isolated or bullied by typically developing peers.

E014: If the other kids will not understand the children with limitations, they will abuse them. (caregiver)

**Subtheme 4.3.2: Family socio-cultural context.** Finally, stakeholders reported lack of awareness and negative beliefs within families. Internalised stigma and the belief that children with DD cannot learn can lead caregivers to "withhold their slowly developing children from school" (A022, mainstream class teacher).

**Theme 5: Political context.**    *Subtheme 5.1*: *Macro political context*. The limited awareness in the community may also be reflected within Ethiopian institutions, as some political figures "don't know what autism is. They don't know the magnitude of the problem. They don't know what to do." (E008, consultant). Additionally, actions may prioritise the education of typically developing children.

E011: When you tell them that [. . .] they need to be inclusive for kids with intellectual disability and so on, they will tell you that "We're not even providing education for the typical ones". . . which is very disappointing that they are, they think that providing education for the typicals [typically developing children] is more important than providing education for the kids with intellectual disability. (clinician)

As such, stakeholders called for the government's attention and commitment to inclusive education, and to services for children with DD more generally. A030 (government official) pointed out that "politics and education should be separated to get out of this problem", to highlight that the commitment to improve education services should not depend on the political beliefs of a given government.

Nonetheless, stakeholders also recognised progress towards building an inclusive society. A019 (academic) and A030 (government official) highlighted how inclusion is a historical process: from initial full exclusion from the society, inclusion is progressing slowly but constantly.

*Subtheme 5.2*: *Meso political context*. While the government's commitment is necessary to guide country-level progress, the will of each principal is key for ground-level implementation.

A022: The most important thing is a positive attitude from the local community and the school leadership. (mainstream class teacher)

On the other hand, principals' negative attitudes can cause rejection of children with DD from schools, or a negative environment towards not only children, but even their special education teachers.

A034: Special needs education teachers tell us they face isolation from mainstream classroom teachers. [. . .] The mainstream teachers themselves look down on special needs education teachers. (special school teacher)

**Theme 6: Legal context.** *Subtheme 6.1*: *Macro legal context*. Professional stakeholders highlighted the importance of national laws for the education of children with DD. They lamented the existence of fragmented, inconsistent and sometimes outdated policies instead of a cohesive disability policy. Similarly, they reported that the Education and Training policy [29] was outdated and had little focus on children with disabilities.

Nonetheless, Ethiopia has ratified international policies such as the Salamanca Statement [30] and the UN CRPD [13] and subsequently issued national action plans. Such steps reassured professionals of the political commitment to positive change.

A030: The Education and Training policy of Ethiopia was issued in 1994 G.C. before Ethiopia adopted the Salamanca statement, it doesn't include the statement. The policy doesn't mention anything about inclusive education. However, article 9 of the Ethiopian constitution states that any foreign laws, regulations, and agreements accepted by Ethiopia will be a part of this constitution. (government official)

Several stakeholders stressed that schools are not legally allowed to reject enrollment of children with disabilities. Another positive policy initiative was the establishment of resource centers promoting special needs education in satellite schools through itinerant teachers tasked with supporting other teachers.

However, stakeholders also reflected on some legal systems that pose barriers to the implementation of such action plans. For example, the relative autonomy of regional governments limits the relevance of national policies. Moreover, teachers have little ability to adapt their teaching and dedicate time to children with DD in mainstream classes, due to the limited flexibility of the national curriculum and the need to teach it in time to prepare students for national exams.

*Subtheme 6.2*: *Meso legal context*. Another reported barrier to inclusion is a policy often adopted by schools, whereby children with DD are only accepted after the family has presented documentation on their disability. Given the challenges in accessing healthcare services, this policy could further delay or stop enrolment.

A026: They refused to accept him if I couldn't bring his medical report. (caregiver)

**Theme 7: Ethical context.** Stakeholders discussed the value of education for children's wellbeing, development, and future and its potential to make children with DD independent and productive to the best of their abilities.

A023: These children will serve their country in the future. [. . .] If you don't teach them, they will become beggars. (mainstream class teacher)

In terms of principles, several professional stakeholders stressed that education was a human right for all children. A minority of them also believed that all children with disabilities had the right to be fully included in mainstream classes and have their needs addressed through inclusive education.

**Theme 8: Setting.** We identified the setting—the physical structure where the intervention takes place—as the school building. Stakeholders reported that most Ethiopian schools

are not accessible for children with disabilities. They called for improved infrastructure, ranging from ramps for children with wheelchairs and accessible toilets, to adequate lighting, removal of haphazard from classrooms and playgrounds, and classroom setups which allow for movement and different activities.

E007: Not only always teaching on the chair and table [is best], so maybe they can learn on the the floor, so setting up the floor also is good. (private special school representative)

Haphazard and ineffective school fencing led to safety concerns in inclusive school settings. Such worries discouraged caregivers from seeking enrolment in inclusive schools, where supervision is also perceived as more loose than in special education settings. On their part, principals and teachers were perceived to be often unwilling to take on children with DD and take responsibility for the risks.

A020: He is not the kind of child that you can manage. What if they enroll him in the school and he escapes their watch and gets hit by a car? (caregiver)

## Discussion

In this study, we interviewed 39 stakeholders and mapped their reports to domains in the Context and Setting dimensions of the CICI framework [25], to explore contextual factors relevant to including children with DD in Ethiopian mainstream schools, with a focus on Addis Ababa. Our data showed that, while general awareness of DD may have increased compared to previous evidence [7, 8], children with DD are still largely excluded from education. Several contextual barriers reported from specific school settings and family contexts up to the the national level were comparable to those highlighted in other African countries [22]. In Theme 1 (Geographical context), we discussed reported challenges with reaching inclusive schools, especially but not only for families living in rural areas, due to their distance and the lack of inclusive public transport. In the first subtheme (Diversity of needs) of Theme 2 (Epidemiological context) we reported stakeholders' discourses on the diverse range of needs and strengths among children with DD and the importance of recognising their potentials. The second subtheme (Gender and DD) concerned child features that may affect their inclusion. Specifically, a small but substantial minority of stakeholders believed that girls with DD may be more excluded than boys, due to the inferior value placed on girls' education in general and to sexual abuse concerns. In Theme 3 (Socio-economic context), we presented stakeholders' reports of various economic and service access challenges that from the macro, or national, level—specifically Ethiopia's low-income status and reliance on external donors—reflect on the meso level—with a lack of material and human resources in schools—and the micro level—with the poverty of many families of children with DD and ultimately the lack of access to individualised support for children themselves. Theme 4 (Socio-cultural context) discusses attitudinal barriers to the inclusion of children with DD, specifically the limited awareness and stigma nationally, among principals, teachers, typically developing children and their parents within school communities and even internalised within families of children with DD. Socio-cultural barriers were reported as decreasing, but still key, and bullying among children was especially a concern for inclusion. In Theme 5 (Political context), the first subtheme (Macro political context) concerns stakeholders' calls for greater political commitment at the national level. The second subtheme (Meso political context) discusses the importance of school principals' commitment to inclusion, equally reported as crucial by stakeholders. Similarly, in Theme 6 (Legal context), the first subtheme (Macro legal context) concerns national policies and the second (Meso legal context) school regulations. Stakeholders described national policies as fragmented, inconsistent and sometimes outdated, while also praising the ratification of relevant international policies and subsequent national action plans. A reported barrier in school regulations is the

requirement for a disability certificate for school enrolment, which may delay enrolment itself. Theme 7 (Ethical context) describes relevant ethical values in Ethiopia, specifically the value placed on the education of all children as a human right and to promote their future contribution to the society. Finally, in Theme 8 (Setting) we discussed school building features as the implementation setting for inclusion, and highlighted stakeholders' concerns for the scarce accessibility and safety of most school buildings.

In practice, as suggested by previous findings, our findings highlight that the availability of special and mainstream schools that accept children with DD is overall limited [4]. This is in contrast to the recognition of the right of children with DD to inclusion in education, highlighted by various professionals in our study referring to both moral and legal frameworks. The latter, for example the Education Sector Development Programmes [12, 31] were praised as showing the government's commitment to inclusion. However, some stakeholders also reported the lack of focus on children with disabilities in the national Education and Training Policy [29]. In mainstream schools, limited awareness among staff and communities at large plays a role in enrolment rejections of children with DD. Stakeholders reported a substantial increase in community awareness of DD in recent years and multiple examples of small- and large-scale awareness-raising efforts. However, our findings show a remaining need to reduce negative attitudes towards children with DD and raising awareness of their ability to learn. Moreover, we identified geographical accessibility as an important barrier: the few special and mainstream schools accepting children with DD often serve large areas, with families coming from far away and struggling with the economic and time costs of transportation. These challenges enhance the burden of care placed on caregiver, often isolated mothers [6, 9, 32]. On the other end, when these struggles are removed, being able to take their child to school reduces the burden on the caregiver, as the school shares the caregiving responsibility. Consequently, making each local school an appropriate setting for the enrolment of children with DD would ensure that they can access education with few struggles, upholding their rights and supporting their caregivers.

Stakeholders also discussed several systematic barriers previously highlighted with regards to including children with DD in mainstream schools in other African countries [22] and children with disabilities more generally in Ethiopia [18–20, 33]: limited material and human resources, lack of specialised training and teacher support, limited curriculum flexibility, and poorly accessible infrastructure. Burningham, Chen, Genovesi et al. [11] analyse in-depth how these barriers can negatively affect children in DD in mainstream schools, causing concerns about their development and learning as well as their safety. In the current analysis we have reported that these barriers also act prior to children with DD being enrolled in schools. For example, teachers who have to manage classes of up to 80 children may reject enrolment of children with DD because they feel unable to take responsibility for their supervision and safety, or to appropriately support their learning. Their limited training in special needs and duty to teach a rigid curriculum further lower their confidence in their ability to meet the needs of children with DD. Therefore, an implementation plan for including children with DD must address these barriers, so that children with DD can both access school and benefit from it.

## Limitations

Eighteen of the 39 interviews were conducted by the first author in English, which was not the native language of the respondents. Interviews in English allowed the first author to immerse herself in the context and gain a deeper understanding on the data. The resulting reports may slightly diverge from the ones that participants would have provided in their mother tongue—

Amharic. However, all the 18 participants were professionals with good conversational English skills and the impact of this limitation on the findings is unlikely to be substantial. The remaining interviews were conducted in Amharic by native Amharic speakers.

Additionally, children with DD who are identified in Ethiopia are typically those with high support needs and substantial cognitive and communication difficulties and we were not able to interview them directly. While caregivers were asked about their children's experiences and some professional participants had lived experience of disability, the fact that children with DD themselves were not interviewed is a limitation of this study.

## Conclusions

Political, legal, and moral frameworks in Ethiopia recognise all children's right to education. However, children with DD currently have limited opportunities to access education in Addis Ababa, either in special schools or the few mainstream schools that accept their enrolment, with even more limitations in other areas of Ethiopia where such services are not available. Sociocultural and socioeconomic barriers, including limited awareness of DD and limited material, human and training resources, hinder their inclusion. Our systematic, in-depth analysis of contextual and setting factors can form the basis for development of an implementation model that addresses such barriers and capitalises on existing facilitators.

## Supporting information

**S1 File. Topic guides.** Topic Guides Used For Interviews.
(DOCX)

**S2 File. Codebook.** Summary of codes developed and applied in the analysis.
(DOCX)

## Acknowledgments

We would like to thank all participants. We also thank Mr Mersha Kinfe and staff at the Addis Ababa University Center for Innovative Drug Development and Therapeutic Trials for Africa (CDT-Africa) for their support in data collection.

## Author Contributions

**Conceptualization:** Elisa Genovesi, Ikram Ahmed, Moges Ayele, Fikirte Girma, Charlotte Hanlon, Rosa Anna Hoekstra.

**Data curation:** Elisa Genovesi, Olivia Burningham, Amanda Chen.

**Formal analysis:** Elisa Genovesi, Ikram Ahmed, Moges Ayele, Winini Belay, Olivia Burningham, Amanda Chen, Fikirte Girma, Liya Tesfaye Lakew.

**Funding acquisition:** Elisa Genovesi, Charlotte Hanlon, Rosa Anna Hoekstra.

**Investigation:** Elisa Genovesi, Ikram Ahmed, Winini Belay, Olivia Burningham, Amanda Chen.

**Methodology:** Elisa Genovesi.

**Project administration:** Elisa Genovesi.

**Software:** Elisa Genovesi.

**Supervision:** Elisa Genovesi, Charlotte Hanlon, Rosa Anna Hoekstra.

**Writing – original draft:** Elisa Genovesi.

**Writing – review & editing:** Elisa Genovesi, Ikram Ahmed, Moges Ayele, Winini Belay, Olivia Burningham, Amanda Chen, Fikirte Girma, Liya Tesfaye Lakew, Charlotte Hanlon, Rosa Anna Hoekstra.

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
