## [Decision Letter · Decision Letter 0]

25 Jan 2024

PONE-D-23-36308Exploring context for implementation of inclusive education for children with developmental disabilities in mainstream primary schools in EthiopiaPLOS ONE

Dear Dr. Genovesi,

Thank you for submitting your manuscript to PLOS ONE. After careful consideration, we feel that it has merit but does not fully meet PLOS ONE’s publication criteria as it currently stands. Therefore, we invite you to submit a revised version of the manuscript that addresses the points raised during the review process.

We look forward to receiving your revised manuscript.

Kind regards,

Mulu Tiruneh

Academic Editor

PLOS ONE

Journal Requirements:

The project has been funded by the UKRI Economic and Social Research Council through a London Interdisciplinary Social Science Doctoral Training Partnership (LISS DTP) studentship (ES/P000703/1), by the British Institute for Eastern Africa through a Thematic Grant and by undergraduate research project funding from King’s College London. RAH, CH, FG and WB receive support from the National Institute for Health and care Research (NIHR200842) using UK aid from the UK Government and CH through the NIHR Global Health Research Group on Homelessness and Mental Health in Africa (NIHR134325) using UK aid from the UK Government. The views expressed in this publication are those of the authors and not necessarily those of the NIHR or the Department of Health and Social Care or other funders. CH receives support from the Wellcome Trust through grants 222154/Z20/Z and 223615/Z/21/Z.

5. We noted in your submission details that a portion of your manuscript may have been presented or published elsewhere. The results in this manuscript have not been published, nor are under consideration anywhere else, however there are two additional studies using the same dataset for two clearly distinct analyses. The study produced a large dataset, as topic guides for stakeholder interviews explored three different areas: (a) stakeholders own views of how inclusive education could benefit or disadvantage children with developmental disabilities in Ethiopia, (b) the current education provision for children with developmental disabilities in the country and barriers and facilitators for inclusion, including for teachers, principals, caregivers and other stakeholders, (c) recommended implementation processes and strategies for effective inclusion of children with developmental disabilities in mainstream schools. These three distinct research questions were answered through three analyses with theme development processes independent from each other.

The enclosed manuscript reports analysis and its companion manuscript that reports analysis will be submitted to PLOS ONE shortly after this. The two are companions as they both use the same implementation framework for theme development, however the enclosed manuscript focuses on the Context and Setting domains only and the other on Implementation only. As such, results are complementary but often refer to different data extracts and are entirely relevant to different topics, rather than two parts of the same manuscript. The two manuscripts can be read independently from each other.

The manuscript reporting analysis, under consideration elsewhere, focuses exclusively on outcomes for children with developmental disabilities, ignoring broader contextual and implementation factors and processes relevant to effective inclusion. Such manuscript represents a background analyses to the enclosed manuscript: it reports that stakeholders consider inclusion in mainstream schools as a potentially beneficial practice for children with developmental disabilities. Such finding is not directly relevant to the research question of analysis, and rather is an important background consideration prior to exploring the readiness of the context for implementing such change, as the enclosed manuscript aims to do. Please clarify whether this [conference proceeding or publication] was peer-reviewed and formally published. If this work was previously peer-reviewed and published, in the cover letter please provide the reason that this work does not constitute dual publication and should be included in the current manuscript.

6. In the online submission form, you indicated that Data will be available upon request. The authors have carefully considered making the data available through a repository. However, in qualitative research and in global health this poses a series of issues, also recognised by evidence (e.g. Tsai et al., 2016). As such, the authors have agreed that it would be unethical to make data publicly available. First, quotes is the manuscript are carefully selected so as to not identify participants or link them to their personal data, but full transcripts may identify them, as professionals working on developmental disabilities in Addis Ababa are few and known to the population. The trascripts may also include personal data beyond participants' perspectives, such as their diagnosis or that of their children. Secondly, the data would largely not be available to researchers in Africa, who are disadvantaged when it comes to accessing publicly available data. Instead, it may be accessible to researchers in high-income countries who may be unfamiliar with the context, leading to potentially harmful interpretations of the data.

7. We note that you have referenced Tesfaye A. (2005).  which has currently not yet been accepted for publication. Please remove this from your References and amend this to state in the body of your manuscript: (Tesfaye A. (2005). [Unpublished]”) as detailed online in our guide for authors

8. We note that you have referenced UN Committee on the Rights of Persons with Disabilities (Sixteenth session). (2016a) which has currently not yet been accepted for publication. Please remove this from your References and amend this to state in the body of your manuscript: (UN Committee on the Rights of Persons with Disabilities (Sixteenth session). (2016a). [Submitted]”) as detailed online in our guide for authors

9. We note that you have referenced UN Committee on the Rights of Persons with Disabilities (Sixteenth session). (2016b) which has currently not yet been accepted for publication. Please remove this from your References and amend this to state in the body of your manuscript: (UN Committee on the Rights of Persons with Disabilities (Sixteenth session). (2016b). [Submitted]”) as detailed online in our guide for authors

Reviewers' comments:

Reviewer's Responses to Questions

**Comments to the Author**

1. Is the manuscript technically sound, and do the data support the conclusions?

Reviewer #1: Yes

Reviewer #2: Partly

2. Has the statistical analysis been performed appropriately and rigorously? 

Reviewer #1: N/A

Reviewer #2: I Don't Know

3. Have the authors made all data underlying the findings in their manuscript fully available?

Reviewer #1: Yes

Reviewer #2: No

4. Is the manuscript presented in an intelligible fashion and written in standard English?

Reviewer #1: Yes

Reviewer #2: No

5. Review Comments to the Author

Reviewer #1: It is very interesting article.I like so much! The manuscript is technically sound able and it has great potential. As is is qualitative study I didn't expect special statistical techniques and it is written in a well organized way.

Reviewer #2: Thank you for the opportunity to review this manuscript. I've included some suggestions and comments.

General comments:

1. Please use Line Numbers next time to facilitate the review.

2. Please ensure that the headings and subheadings are numbered consistently. For example, Materials and Methods section were no numbered. Some numbers of subheading are duplicated (e.g. 1.3.3.1).

3. It is recommended to upload the interview guideline and data analysis (coding).

Specific comments:

A. Title:

1. Authors should use ‘Addis Ababa, Ethiopia’ instead of ‘Ethiopia’ only which may limit the generalizability of the findings.

B. Introduction:

1. Could you please let us know the latest statistical data on the number of children with DD in Ethiopia?

2. In the authors' second paragraph, they stated that services for children with developmental disabilities (DD) are limited, and that there are unmet needs in accessing professional support for these children. Can you first provide an example of the types of services and professional support that are DD children need?

C. Materials and Methods

1. It is ‘Materials and Methods’ not ‘Methods’, based on guidelines.

2. I wonder how semi-structured interviews guideline was developed, validated, and relied?

3. This part should be in Materials and Methods section not in results.

‘Analysis: Below we report our analysis of factors relevant to the Context and Setting dimensions of the CICI framework, for the implementation of interventions to include children with DD in mainstream schools, based on interview data. Quotes are reported verbatim, alongside the participant’s code created from a letter (“E” for interviews in English, “A” for interviews in Amharic) and a unique number’ .

D. Results

1. I would prefer to summarise your participants in a table then report it.

E. Discussion:

1. It wasn't enough discussion. Further divided into concise subsections based on outcomes.

F. References:

1. Please use the right style of in-text referencing and bibliography.

6. PLOS authors have the option to publish the peer review history of their article (what does this mean?). If published, this will include your full peer review and any attached files.

Reviewer #1: No

Reviewer #2: No

---

## [Author Response · Author response to Decision Letter 0]

5 Apr 2024

Please see the point-by-point response in the attached Response to Reviewers

---

## [Editor Report · Decision Letter 1]

16 May 2024

PONE-D-23-36308R1Exploring context for implementation of inclusive education for children with developmental disabilities in mainstream primary schools in EthiopiaPLOS ONE

Dear Dr. %LAST_NMAE%,

Thank you for submitting your manuscript to PLOS ONE. After careful consideration, we feel that it has merit but does not fully meet PLOS ONE’s publication criteria as it currently stands. Therefore, we invite you to submit a revised version of the manuscript that addresses the points raised during the review process.

We look forward to receiving your revised manuscript.

Kind regards,

Mulu Tiruneh

Academic Editor

PLOS ONE

---

## [Author Response · Author response to Decision Letter 1]

27 Jun 2024

Dear Dr Mulu Tiruneh,

Thank you for kindly advancing the review process for our manuscript "Exploring context for implementation of inclusive education for children with developmental disabilities in mainstream primary schools in Ethiopia" and conveying the comment offered by Reviewer 1 in this second round of reviews.

In the manuscript we are now submitting, we have simplified language at lines 190-192, based on the reviewer’s comment that there was “No need of saying in such scientific writing”, which we agree with and appreciate.

We hope that the revision is to your satisfaction and look forward to hearing from you.

Yours sincerely, on behalf of all co-authors,

Elisa Genovesi

---

## [Editor Report · Decision Letter 2]

9 Jul 2024

Exploring context for implementation of inclusive education for children with developmental disabilities in mainstream primary schools in Ethiopia

PONE-D-23-36308R2

Dear Dr. Genovesi,

We’re pleased to inform you that your manuscript has been judged scientifically suitable for publication and will be formally accepted for publication once it meets all outstanding technical requirements.

Kind regards,

Mulu Tiruneh

Academic Editor

PLOS ONE
---

## [Editor Report · Acceptance letter]

31 Jul 2024

PONE-D-23-36308R2 

PLOS ONE

Dear Dr. Genovesi, 

I'm pleased to inform you that your manuscript has been deemed suitable for publication in PLOS ONE. Congratulations! Your manuscript is now being handed over to our production team.

Kind regards, 

on behalf of

Mr. Mulu Tiruneh 

Academic Editor

PLOS ONE